# Long-Term Retrogradation Properties and In Vitro Digestibility of Waxy Rice Starch Modified with Pectin

**DOI:** 10.3390/foods12213981

**Published:** 2023-10-31

**Authors:** Yuheng Zhai, Hao Zhang, Jiali Xing, Shangyuan Sang, Xinyan Zhan, Yanan Liu, Lingling Jia, Jian Li, Xiaohu Luo

**Affiliations:** 1National Engineering Research Center for Cereal Fermentation and Food Biomanufacturing, Jiangnan University, Wuxi 214122, China; 6190112167@stu.jiangnan.edu.cn (Y.Z.); leslie198907@foxmail.com (H.Z.); 2Ningbo Academy of Product and Food Quality Inspection (Ningbo Fibre Inspection Institute), Ningbo 315048, China; hellojiali77@gmail.com; 3Zhejiang-Malaysia Joint Research Laboratory for Agricultural Product Processing and Nutrition, Key Laboratory of Animal Protein Food Deep Processing Technology of Zhejiang Province, College of Food and Pharmaceutical Sciences, Ningbo University, Ningbo 315832, Chinaliuyanan@nbu.edu.cn (Y.L.); jialingling@nbu.edu.cn (L.J.); 4Key Laboratory of Green and Low-Carbon Processing Technology for Plant-Based Food of China National Light Industry Council, Beijing Technology and Business University, Beijing 100048, China

**Keywords:** waxy rice starch, pectin, pasting properties, retrogradation properties, in vitro digestibility

## Abstract

In recent years, the blending of hydrocolloids and natural starch to improve the properties of natural starch has become a research hotspot. In this study, the effects of pectin (PEC) on the retrogradation properties and in vitro digestibility of waxy rice starch (WRS) were investigated. The results showed that PEC could significantly (*p* < 0.05) reduce the retrogradation enthalpy and reduce the hardness of WRS gel. X-ray diffraction results indicated that PEC could reduce the relative crystallinity of the composite system, and the higher the PEC content, the lower the relative crystallinity. When the PEC content was 10%, the relative crystallinity of the composite system was only 10.6% after 21 d of cold storage. Fourier transform infrared spectroscopy results proved that the interaction between PEC and WRS was mainly a hydrogen bond interaction. Furthermore, after 21 d of cold storage, the T_23_ free water signal appeared in the natural WRS paste, while only a small free water signal appeared in the compound system with 2% PEC addition. Moreover, addition of PEC could reduce the starch digestion rate and digestibility. When the content of PEC increased from 0% to 10%, the digestibility decreased from 82.31% to 71.84%. This study provides a theoretical basis for the further application of hydrocolloids in starch-based foods.

## 1. Introduction

Waxy rice is an important food crop in Asia, especially in China [1]. It has high nutritional value and has the effects of strengthening the spleen and stomach, tonifying Qi, and stopping diarrhea. Waxy rice starch (WRS) is the essential component of waxy rice, which is widely used in the food industry. The main component of WRS is amylopectin (more than 95%). However, in the process of production, circulation, and storage, WRS-based food will undergo multiple freeze–thaw processes due to temperature fluctuations. Retrogradation consists of two stages: a short- and a long-term retrogradation. The long-term retrogradation of starch is mainly caused by the recrystallization of amylopectin, which takes a long time and has thermal reversibility [2]. In the process of freeze–thaw, natural WRS will experience retrogradation, poor mechanical stability and poor storage stability, which affects the quality of WRS-based food. Waxy starch also usually shows higher digestibility than normal starch. Raigond et al. [3] have proved that long-term consumption of high levels of rapidly digestible starch (RDS) foods increases the risk of diseases, such as obesity and diabetes. However, there is still an urgent need to further reduce the digestibility of waxy rice starch so it can be used as an ingredient in some low glycemic index (GI) foods [4]. Although chemical modification [5] and enzymatic modification [2] are commonly used to improve the properties of waxy rice starch in food processing, chemical modification has poor safety and environmental protection, and enzymatic modification is expensive. Therefore, a green, safe, and economic way to improve the defects of WRS is needed.

Hydrocolloids, such as sodium alginate, guar, and xanthan gum, are rich in hydrophilic groups and have good water binding ability [6]. Studies have shown that the combination of hydrocolloid and starch can improve the gelatinization, rheology, retrogradation, and digestion of starch to a certain extent, and maintain the overall quality of the final product [7]. Lin et al. [8] found that the addition of xanthan gum and konjac gum had a great influence on the gelatinization and rheology of mung bean resistant starch. Compared with pure resistant starch gel, the gelatinization and rheological properties of mung bean resistant starch/hydrophilic colloid composite paste were improved. Dobosz et al. [9] studied the effects of xanthan gum on short-term and long-term retrogradation of potato starch gels with different amylose/amylopectin ratios and found that the presence of xanthan gum promoted the short-term retrogradation of samples after preparation. At the same time, the presence of xanthan gum hindered the long-term aging of potato starch gels with high amylopectin content but promoted the long-term aging of ordinary potato starch gels. Wang at al. [10] found that the addition of *Artemisia sphaerocephala Krasch* gum (ASKG) apparently reduced the digestibility of waxy rice starch. When the addition of ASKG was increased from 0 to 15% (*w*/*w*), the RDS of WRS samples decreased from 54.9% to 32.9%, respectively, and the RS increased from 28.2% to 38.4%, respectively. 

Pectin (PEC) is a hydrophilic polysaccharide extracted from higher plant tissues, such as roots, stems, and leaves, which is an excellent hydrocolloid. In the food industry, PEC can be added to food, which plays a role in gelation, thickening and improving the texture of food, and is an excellent food additive. In addition, scholars have found that PEC could influence the gelatinization, retrogradation, and digestion properties of starch [11]. Xie et al. [12] found that the presence of PEC significantly changed the viscoelasticity of corn starch gel (*p* < 0.05) and delayed the retrogradation of starch gel. Zhang et al. [13] added PEC to corn starch and studied the effect of PEC molecular weight on retrogradation and digestion of starch. It was found that, with the increase of PEC molecular weight, the relative crystallinity of starch-PEC mixture decreased significantly, and the proportion of slowly digestible starch (SDS) and resistant starch (RS) in corn starch increased significantly (*p* < 0.05). However, there is still a lack of relevant data on the effect of PEC on the functional properties, long-term retrogradation and digestibility of WRS.

Our previous work found that PEC could influence the gelatinization and freeze–thaw stability of WRS [14]. Thus, the objective of the present study was to evaluate the effects of PEC at different addition on the long-term retrogradation and in vitro digestibility of WRS. This study is of great significance for further understanding the effect and interaction between hydrocolloids and starches. It may also provide a new idea regarding how to produce long-term storage and low GI starch-based food products.

## 2. Materials and Methods

### 2.1. Materials

WRS was extracted from milled waxy rice flour which was purchased from Kemen Noodle Manufacturing Co., Ltd. (Changsha, China). The amylose content of the WRS was 1.14%. PEC extracted from apples was purchased from Youbaojia Food Co., Ltd. (Zhengzhou, China). The degree of esterification of PEC was 13.5%. Isoamylase (Cas: 9067-73-6), pancreatic-amylase (Cas: 9000-90-2), amyloglucosidase (Cas: 9032-08-0) from Aspergillus niger, and glucose oxidase/peroxidase reagent (GOPOD) were all purchased from Megazyme (Wicklow, Ireland). All other chemical and reagents were of analytical grade (Sinopharm Chemical Reagents Co., Ltd., Shanghai, China).

### 2.2. Preparation of WRS−PEC Complexes

The WRS−PEC complexes were prepared in accordance with the method of Lin et al. [8], with some modification. Briefly, WRS (8%, *w*/*v*, based on dry basis) was dispersed into distilled water, and PEC with mass of 0%, 2%, 4%, 6%, 8%, and 10% (*w*/*w*, based on dry basis of starch) was added to form the mixtures. The mixtures were mixed and stirred at 100 °C for 30 min until they were fully gelatinized. The starch pastes were then cooled to room temperature and transferred to the sealing bags, and the sealing bags were stored at 4 °C in the refrigerator for 0, 7, 14, and 21 days, respectively.

### 2.3. Thermal Properties

The thermal properties of the WRS−PEC complexes were determined using a differential scanning calorimeter (DSC3, Mettler Toledo, Zurich, Switzerland) according to the method of Chen et al. [15], with some modifications. WRS containing 0%, 2%, 4%, 6%, and 10% PEC (*w*/*w*, based on starch weight) were prepared. A total of 3 mg of prepared samples were mixed with 6 μL of deionized water and hermetically sealed in an aluminum pan. The samples were equilibrated at 4 °C for 12 h and heated from 30 °C to 100 °C at a rate of 10 °C/min. Subsequently, the gelatinized samples were cooled to room temperature and stored at 4 °C for 7, 14, and 21 days to perform the retrogradation process. Then, DSC thermal scanning was conducted according to the same heating procedure described above to test the retrogradation enthalpy of the samples (Δ*H_r_*).

### 2.4. Textural Properties

The textural properties of WRS and WRS−PEC complexes were investigated by TA-XT plus texture analyzer (Stable Micro Systems, Godalming, UK) according to the previous method with some modifications [13]. The samples obtained from Section 2.2 were immediately transferred to plastic cups with covers (30 mm diameter, 20 mm height) and the texture profile analysis of pastes were performed by using a 25 mm diameter cylindrical probe which punctured the gels to 10 mm depth at a pre and post speed of 1 mm/s. The hardness of the samples was reported.

### 2.5. X-ray Diffraction (XRD) Analysis

The WRS gels with or without PEC were immediately freeze-dried and stored at 4 °C for 7, 14, and 21 days, then milled to pass through a 100-mesh sieve. X-ray diffraction analysis of samples was performed using a D2 Phaser X-ray diffractometer (Bruker, Krlsruhe, Germany) operating at 45 kV and 30 mA with Cu Kα radiation (λ = 1.54 Å) according to the previous method with some modification [16]. The samples were scanned from 4° to 40° at a scan rate of 4°/min. The data were analyzed using Jade 6.0 software (Materials Data Inc., Livermore, CA, USA), and the degree of crystallinity (*R_c_*) was calculated as follows:(1)$$R_{c}(\%)=\frac{A_{c}}{A_{c}+A_{a}}\times 100$$
where *A_c_* and *A_a_* are the crystalline area and the amorphous area of the XRD patterns, respectively.

### 2.6. Fourier Transform Infrared (FTIR) Spectroscopy

FTIR spectra of the freeze-dried WRS−PEC complex powders were obtained on a PerkinElmer spectrometer fitted with an ATR accessory (PerkinElmer, MA, USA). The spectra were recorded from 4000 to 400 cm^−1^ by accumulating 64 scans at a resolution of 4 cm^−1^. The infrared spectra were analyzed by Omnic software 8.2 (Thermo Fisher Scientific, Inc., Waltham, MA, USA), and the absorbance values at wavelengths of 1047 cm^−1^, 1022 cm^−1^, and 995 cm^−1^ were recorded [17].

### 2.7. Low-Field Nuclear Magnetic Resonance (LF-NMR)

The water distribution and migration of the WRS−PEC complexes samples stored at 4 °C for 7, 14, and 21 days were measured using an NMR analyzer (MesoMR23-060V-I, Suzhou Niumag Analytical Instrument Corporation, Suzhou, China) according to the previous method [18]. The resonance center frequency was first adjusted with the FID sequence, and then the data were collected using the Carr–Purcell–Meiboom–Gill (CPMG) pulse sequence. For system CPMG sequence parameters: The number of sampling points, echo number, and echo time were 1,000,056, 4000, and 0.50 ms respectively. The samples were scanned eight times, and the frequency of the repeat sweep was 200 kHz.

### 2.8. In Vitro Digestibility

The in vitro digestibility of complexes was determined based on the Englyst method, with slight modifications [19]. Briefly, the WRS−PEC complexes (containing 100 mg starch, dry basis) were added to a screw cap tube and supplied with 5 mL of the digestive juice. Digestive juice was prepared using the method by Pan et al. [20]. Briefly, amounts of α-amylase and amyloglucosidase were dissolved in 0.1 M sodium acetate buffer (with 4 mM CaCl_2_, pH 5.2) to a final concentration of 290 units/mL and 15 units/mL, respectively. The tube was incubated in a 37 °C shaking water bath with a speed of 160 r/min. At designated times (5, 10, 15, 20, 40, 60, 80, 120, 160, and 200 min), aliquots (50 μL) of samples were taken into a 2 mL microcentrifuge tubes and mixed with 450 μL of absolute ethanol to deactivate digestive enzymes, followed by centrifugation at 10,000× *g* for 2 min. Then, 0.1 mL of D-glucose standards and sample supernatant containing released D-glucose was added to 3.0 mL of GOPOD reagent and incubated at 40 to 50 °C for 20 min. All absorbances were measured at 510 nm against the reagent blank to calculate the concentration of glucose at different intervals. The percentages of rapidly digestible starch (RDS), slowly digestible starch (SDS), and resistant starch (RS) of each starch sample were calculated as follows:(2)$$\mathrm{RDS}(\%)=\frac{G_{20}\times 0.9}{M_{0}}\times 100$$
(3)$$\mathrm{SDS}(\%)=\frac{{(G}_{120}-G_{20})\times 0.9}{M_{0}}\times 100$$
(4)RS(%)=100−RDS−SDS
where M_0_ represents the initial sample mass (mg), G_20_ represents the contents of glucose released after 20 min, and G_120_ represents the contents of glucose released after 120 min, respectively.

The digestion process was fitted with a first-order kinetic equation, where *C* is the starch digestibility at time *t*, *C_∞_* is the total starch digestibility at the end point, and *k* is the digestion rate constant.
(5)$$C=C_{\infty }\left(1-e^{-kt}\right)$$

The hydrolysis index (*HI*) was calculated based on dividing the area under the hydrolysis curve (0~180 min) of the sample by the area of a standard material (white bread) over the same period [21]. The estimated glycemic index (*eGI*) was calculated using the equation described by Maibam et al. [22]:(6)eGI=39.71+0.549HI

### 2.9. Statistical Analysis

Each test was determined at least three times, and data were expressed as mean ± standard deviation. Statistical significance was analyzed by one-way analysis of variance (ANOVA) by using SPSS 16.0 (SPSS Inc., Chicago, IL, USA). The value of significance was considered as *p* < 0.05.

## 3. Results

### 3.1. Thermal Properties

The essence of the retrogradation process of starch is that the movement of starch molecules is weakened when the gelatinized starch is cooled, and the amylose and amylopectin molecules in the starch are rearranged in the form of hydrogen bonds, resulting in the recrystallization of starch paste. The amylopectin in WRS is the main component, so the WRS paste will mainly undergo long-term aging. The retrogradation enthalpy (Δ*H_r_*) of starch mainly reflects the degree of recrystallization of amylopectin. Table 1 shows the changes of Δ*H_r_* of WRS samples with different PEC additions during cold storage at 4 °C. It can be seen from the table that with the extension of refrigeration time, Δ*H_r_* of all samples gradually increases, that is, the aging degree of starch increases. After 21 days of cold storage, the Δ*H_r_* of natural glutinous rice starch increased to 6.37 J/g. However, with the increase of PEC content, the Δ*H_r_* of starch at different refrigerated times decreased, and when the pectin content was 10% and refrigerated for 21 days, the Δ*Hr* of the composite system decreased to 4.29 J/g. This indicated that pectin may inhibit the long-term aging of amylopectin in WRS. Chen et al. [15] found that, after 28 days of cold storage, compared with natural rice starch, the addition of pullulan decreased the Δ*H_r_* of starch, and the larger the addition amount was, the lower the Δ*H_r_* of the sample was. Pullulan polysaccharide plays a role in wrapping rice starch, which limits the gelatinization of part of the starch, allowing it to retain many orderly and compact structures in the granules, thereby inhibiting aging. With the increase in PEC content, PEC molecules are wrapped around WRS, competing with starch for water absorption, which limits its gelatinization effect. In the aging process, incomplete gelatinization of starch inhibits amylose leakage, thereby weakening the short-term aging of starch and inhibiting the long-term aging of amylopectin.

### 3.2. Texture Profile Analysis (TPA)

The increase in hardness of starch-based food is one of the most significant manifestations of starch retrogradation [23]. The hardness of the WRS−PEC composite system with different PEC content stored at 4 °C for different times is shown in Figure 1. It can be seen from the figure that the hardness of WRS paste with or without PEC increases with the extension of low-temperature storage time. After 21 days storage, the hardness of natural WRS paste increased from 135.29 g to 168.25 g. However, the addition of PEC significantly (*p* < 0.05) reduced the hardness of starch gel. When the addition of PEC was 10%, the hardness of glutinous rice starch paste decreased from 168.25 g to 113.93 g after 21 days of storage. PEC could enter the inside of starch, increasing the distance between starch molecules, then reducing the number of hydrogen bonds formed between starch molecules. At the same time, PEC might form hydrogen bonds with starch molecules through hydroxyl groups in WRS molecules, interfering with the formation of the starch’s orderly structure, limiting the rearrangement of the starch chain, and inhibiting starch recrystallization. Zhang et al. [13] found that, after adding PEC to corn starch (CS), the gel hardness of CS-PEC complex decreased after 14 days of low-temperature storage compared with natural corn starch, and the adding low ester PEC had a better effect. Similarly, when Xiao et al. [23] added antilisterial grass carp protein hydrolysate, the hardness of starch gel decreased during storage.

### 3.3. XRD Pattern Analysis

As shown in Figure 2E, the natural WRS has a typical A-type crystalline structure with two intense peaks at about 15.1° and 23.0° (2θ) and an unresolved doublet at approximately 17.1° and 18.1° (2θ). During the retrogradation process of starch paste, starch molecules aggregate with each other through hydrogen bonds to form crystals, thereby increasing the crystallinity of starch gel [15]. The X-ray diffraction patterns of WRS−PEC complexes at different storage periods are shown in Figure 2. The relative crystallinity of each sample was labeled one by one. It can be seen from Figure 2A that the crystalline peak type changed compared with the natural WRS. This may because, during the gelatinization process of WRS, the particles continue to absorb water and expand, resulting in the fracture of the intramolecular hydrogen bond in the crystalline region and the destruction of the double helix structure formed by the side chain of amylopectin. There was no obvious diffraction peak in the XRD patterns of WRS and WRS−PEC complex.

The greater the relative crystallinity of starch, the greater the proportion of crystallization area in the sample, that is, the higher the degree of retrogradation and regeneration of the sample. It can be seen from Figure 2 that the relative crystallinity of WRS paste without PEC increases from 10.80% to 17.14% after 21 days of storage. The addition of PEC can significantly reduce the relative crystallinity of WRS−PEC mixed system (*p* < 0.05), and the change of relative crystallinity is negatively related to the amount of PEC. When the amount of PEC is 10%, the relative crystallinity of composite system is only 10.55% after 21 days of refrigeration. This might be because the addition of PEC inhibits the formation of hydrogen bonds between starch molecules and hinders the formation of double helix structure of Starch Chain [24], and PEC can weaken the mobility of water molecules near the starch molecular chain, thus inhibiting the participation of water molecules in the process of starch retrogradation [15]. The above results further show that PEC can inhibit the long-term retrogradation of WRS to a certain extent, which is consistent with the results of DSC and texture experiment.

### 3.4. FTIR Analysis

Fourier infrared spectroscopy could be used to determine whether there is an intermolecular hydrogen bond and the strength of the hydrogen bond in the sample. In this experiment, FTIR spectroscopy is used to investigate the force between PEC and WRS. The FTIR spectra of WRS−PEC complex freeze-dried powder refrigerated at 4 °C for 7 days were shown in Figure 3. It can be seen from the figure that the interaction between pectin and glutinous rice starch is mainly hydrogen bonds. This is because there is no new absorption peak in the compound after adding pectin, so there is no covalent binding between PEC and WRS. Meanwhile, both starch and PEC molecules contain many hydrophilic hydroxyl groups, so there is no hydrophobic interaction between them [25].

As shown in Figure 3, all samples have a strong, wide absorption peak in the range of the 3500~3200 cm^−1^ wave number, which represented the absorption peak of intermolecular hydroxyl stretching vibration [26] and reflects the strength of intermolecular and intramolecular hydrogen bond forces of starch. Due to the large number and different strength of hydrogen bonds in starch, wide and strong characteristic peaks will be formed. As can be seen from the figure, as the amount of PEC added increased from 2% to 10%, the absorption peak number of the complex decreased from 3315 cm^−1^ to 3297 cm^−1^, indicating that the intermolecular hydrogen bond in the complex was gradually strengthened. The intermolecular force between PEC and WRS increased, which inhibited the interaction between starch molecules and slowed down the retrogradation of starch.

The absorption bands at 1047 cm^−1^, 1022 cm^−1^, and 995 cm^−1^ were sensitive to the ordered or crystalline structures, amorphous structures, and double helical structure of starch, respectively [27]. Furthermore, the ratio of absorbances (1047/1022) cm^−1^ was used to quantify the degree of order in starch samples, and the ratio of absorbances (1022/995) cm^−1^ was usually used to assess the interactions between water and starch [28]. As can be seen from Table 2, with the increase of PEC addition, the value of (1045/1022) cm^−1^ decreased significantly (*p* < 0.05) from 0.54 to 0.44, indicating that PEC inhibited the interaction between starch molecules, hindered the rearrangement of starch molecules to form an ordered structure, and then inhibited retrogradation. In addition, as PEC increased from 0% to 10%, the value of (1022/995) cm^−1^ increased significantly (*p* < 0.05) from 0.77 to 0.97, indicating that the addition of PEC enhanced the interaction between water molecules and starch.

### 3.5. Water Mobility and Distribution of WRS−PEC during Cold Storage

Low-field NMR (LF-NMR) technology is a new technology used to study the water distribution and migration in food system in recent years. It can reflect the motion characteristics of water molecules in the reaction system by detecting parameters such as proton relaxation behavior and amplitude [29]. The water mobility and water distribution in different states of starch gel are positively correlated with the short-term and long-term retrogradation of starch during cold storage [30]. The water distribution of WRS−PEC complexes with different PEC additions (0%, 2%, 6%, and 10%, *w*/*w*) during aging was studied by LF-NMR. The spin–spin relaxation time (T_2_) map is shown in Figure 4. According to previous studies, the relaxation time T_2_ can be divided into three parts: T_21_ (<10 ms), T_22_ (10~1000 ms) and T_23_ (>1000 ms), which are related to the bound water, non-flowing water, and free water in the system respectively. The proportion of the peak area corresponding to the characteristic peak of each region is recorded as A_21_, A_22_, and A_23_. The larger the peak area of the characteristic peak, the higher the relative moisture content in this part. The length of relaxation time T_2_ determines the degree of tight binding between water molecules and substances. The more obvious T_2_ was, the closer it binds to substances, and the mobility of water molecules is bound by starch [29].

It can be seen from Figure 4A,B that the starch gel that gelatinized and was refrigerated for 7 days has only two characteristic peaks, corresponding to bound water and non-flowing water respectively. After 14 days of cold storage, the natural WRS starch gel showed a third characteristic peak at 2154.43 ms (Figure 4C), indicating that some free water had been separated out at this time. This is mainly because the water holding capacity of starch gel decreases due to starch retrogradation. After 21 days of cold storage, the glutinous rice starch paste with 2% pectin also showed a third characteristic peak at 1232.85 ms (Figure 4D), and the area of T_23_ characteristic peak of natural glutinous rice starch paste was increasing, indicating that, with the extension of storage time, the retrogradation of starch intensified and the water holding capacity weakened. However, in general, with the increase of PEC addition, T_2_ of the complex shifted to the left, that is, the combination of water and starch was closer. This shows that the addition of pectin can effectively bind the water molecules in the system, reduce the precipitation of free water, and then inhibit the retrogradation of starch.

The water distribution characteristic parameters of glutinous rice starch pectin complex during storage are shown in Table 3. Compared with natural glutinous rice starch paste, T_21_ and T_22_ of starch paste added with PEC are reduced, indicating that pectin limits the fluidity of water molecules to a certain extent. Ge et al. [31] added konjac glucomannan to wheat starch and found that adding konjac glucomannan during storage can bind water molecules in starch, to maintain high water holding capacity of starch. In this study, T_2_ of the sample gradually decreased with the extension of storage time (Table 3). This phenomenon may be due to the formation of ordered structure of starch molecules during retrogradation, resulting in water molecules being fixed in the recrystallized structure. After 7 days of storage, the proportion of A_21_ in starch paste decreased and the proportion of A_22_ increased, which indicates that part of the bound water migrated to non-flowing water during storage, and the migration rate of starch paste with pectin was slow. After storage for 14 days, the A_21_ and A_22_ of natural WRS paste decreased, and the water began to migrate to the free water part with high fluidity. Long-term retrogradation leads to the decrease of starch’s water-holding capacity, and the bound water and non-flowing water are gradually separated from complex macromolecules and transformed into free water. After 21 days of storage, the starch paste with 6% and 10% PEC still has only two signal areas, indicating that the addition of PEC may change the relaxation behavior and migration characteristics of water molecules in the system, strengthening the binding tightness between water molecules and starch, and then inhibiting the long-term retrogradation of starch.

### 3.6. In Vitro Digestibility

The classical Englyst [19] method was used to study the effects of different PEC additions on the digestive characteristics of WRS. The contents of each digestive component of the composite system are shown in Table 4. It can be seen from the table that the contents of rapid digestible starch (RDS), slow digestible starch (SDS), and resistant starch (RS) of gelatinized natural WRS are 64.31%, 20.11%, and 15.75%, respectively. The addition of PEC significantly (*p* < 0.05) decreased the digestibility of WRS. With the increase of PEC content from 2% to 10%, the SDS and RS of the composite system increased from 20.25% and 16.03% to 21.87% and 26.52%, respectively. Meanwhile, the RDS of the composite system decreased significantly. Ma et al. [32] added PEC to corn starch to improve its digestibility. The results showed that the addition of PEC could significantly reduce RDS (*p* < 0.05), and when the addition amount was 10% (*w*/*w*), the RDS of corn starch decreased from 73.1% to 51.5%. Gao et al. [27] found that, when the addition amount of dandelion root polysaccharide (DRP) was 15%, the contents of SDS and RS in corn starch increased from 1.87% and 93.75% to 3.07% and 95.02%, respectively. The addition of DRP reduced the in vitro digestibility of corn starch. The digestibility of starch depends on the interaction of various factors, such as morphology, crystalline structure, and the proportion of amylopectin. In this study, the reduction of RDS in the composite system may occur because PEC would wrap on the surface of glutinous rice starch to prevent the starch from contacting the digestive solution. At the same time, the wrapped PEC will also inhibit the water absorption and expansion of starch, thus reducing the degradation rate of starch. The higher the RDS of starch, the more likely it is to cause chronic diseases, such as postprandial hyperglycemia. RS can promote the balance of intestinal flora and is of great significance in maintaining human intestinal health [2]. Therefore, adding PEC to WRS can improve its digestibility and is of great significance to preventing chronic diseases, such as hyperglycemia and diabetes.

### 3.7. Kinetics of Starch Hydrolysis

The digestion and hydrolysis curve of gelatinized natural WRS and WRS−PEC complexes during simulated small intestinal digestion are shown in Figure 5. It can be seen from the figure that all samples gradually stabilized before and after 120 min. The addition of PEC could affect the hydrolysis rate of WRS. When the amount of PEC was 10% and hydrolyzed for 40 min, the hydrolysis rate of starch decreased from 71.55% to 58.67%. Zhang et al. [13] found the same phenomenon when studying the effect of PEC on the digestibility of corn starch.

In order to evaluate the effect of PEC more accurately on the digestibility of WRS, the first-order equation model was used for nonlinear fitting, and the hydrolysis kinetic parameters were calculated (Table 5). It can be seen from the table that the correlation R^2^ of all data is greater than 0.97, indicating that the hydrolysis first-order equation is well fitted. *C_∞_* value is the percentage of theoretically digested starch at the end of the reaction, which is related to the equilibrium concentration of digestion. The addition of PEC increased from 0% to 10%, and the *C_∞_* value decreased from 82.31% to 71.84%, which is consistent with the research results of Chen et al. [33]. They found that the *C_∞_* value decreased after adding pullulan polysaccharide to rice starch. The *k* value is related to the rate of amylase hydrolysis. The *k* value of natural glutinous rice starch was the highest (0.073). When the amount of PEC was between 2~10%, the *k* value decreased from 0.069 to 0.056. These results showed that the addition of PEC significantly reduced the hydrolysis rate and digestion rate of starch. Meanwhile, with the addition of PEC, the *eGI* of WRS decreased gradually (Table 5). This may occur because the addition of PEC increases the viscosity of starch suspension, which may hinder the diffusion of the enzyme and reduce the rate of enzymatic hydrolysis of starch. Another explanation is that PEC forms PEC layer around starch particles, forming steric hindrance, which limits the contact between hydrolase and starch [32]. In addition, Bai et al. [34] found that the addition of PEC significantly reduced the starch digestion rate due to the interaction between PEC and starch glucosidase.

## 4. Conclusions

In this research, the effects of adding PEC on the retrogradation properties and in vitro digestibility of WRS were investigated. PEC could significantly reduce the retrogradation enthalpy of WRS. The addition of PEC measured by texture analyzer could inhibit starch retrogradation and recrystallization, thereby reducing the hardness of the gel. XRD results showed that the addition of PEC could significantly reduce the relative crystallinity of the composite system, and the higher the PEC content, the lower the relative crystallinity. FTIR results further proved that PEC can effectively inhibit the interaction between starch molecules in the system, reduce the degree of crystallization, and thus achieve the effect of inhibiting the retrogradation of WRS. The results of LF-NMR showed that, after 21 d of cold storage, the T_23_ free water signal appeared in the natural WRS paste, while only a small free water signal appeared in the compound system with 2% PEC addition. Digestion experiments showed that PEC could improve the digestibility of WRS to some extent and reduce the starch digestion rate and digestibility. When the PEC content increased from 0% to 10%, the digestibility decreased from 82.31% to 71.84%. These results are beneficial for the development of WRS-based products with high quality. Furthermore, this study provides a theoretical basis for the potential application of hydrocolloids in improving the processing properties and storage stability of starch-based foods.

## Figures and Tables

**Figure 1 foods-12-03981-f001:**
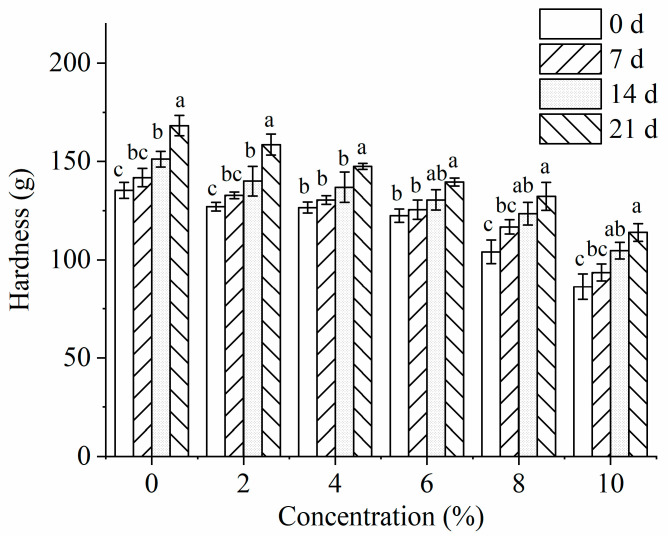
The hardness of WRS−PEC mixed system with different PEC additions during different storage periods (0 d, 7 d, 14 d, and 21 d). Means with different lower case letters superscripts are significantly different at *p* < 0.05.

**Figure 2 foods-12-03981-f002:**
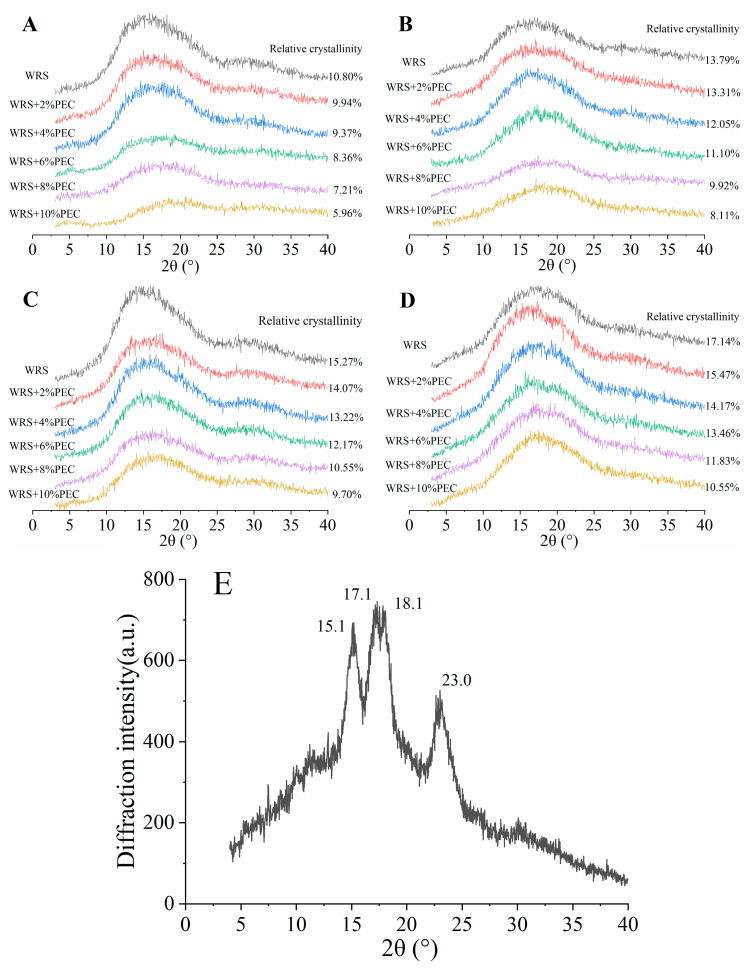
X−ray diffraction patterns of WRS−PEC mixed system in different storage periods (**A**). 0 d; (**B**). 7 d; (**C**). 14 d; (**D**). 21 d and natural WRS (**E**).

**Figure 3 foods-12-03981-f003:**
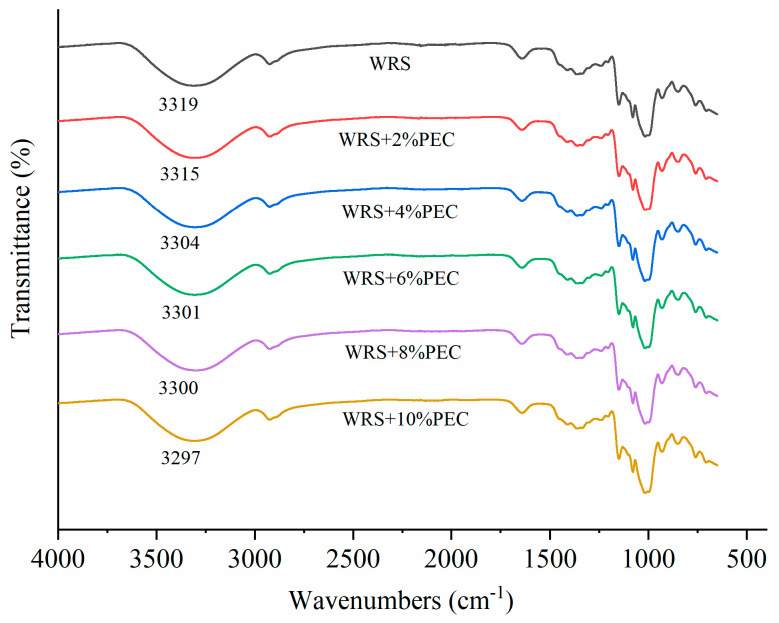
FT−IR spectrogram of WRS−PEC mixed system with different PEC additions (4 °C for 7 days’ storage).

**Figure 4 foods-12-03981-f004:**
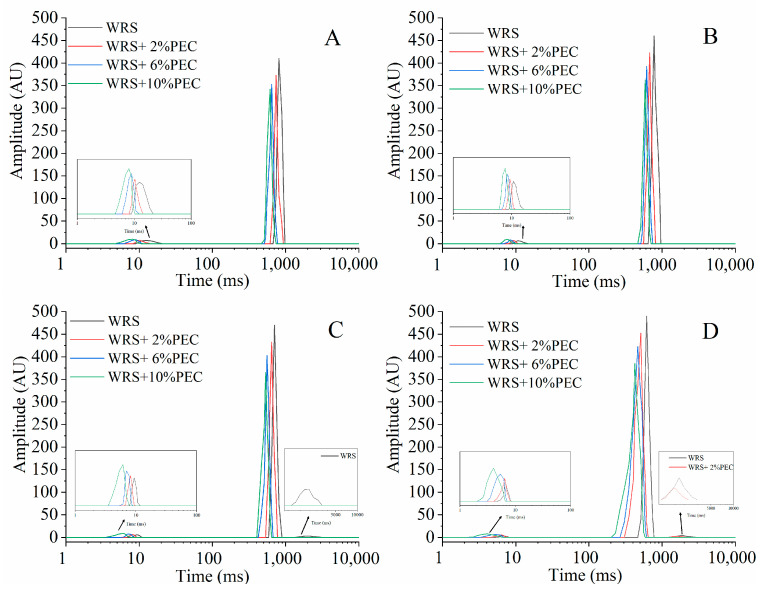
The relaxation time spectrum of LF-NMR for WRS−PEC mixed system during storage (**A**). 0 d; (**B**). 7 d; (**C**). 14 d; (**D**). 21 d.

**Figure 5 foods-12-03981-f005:**
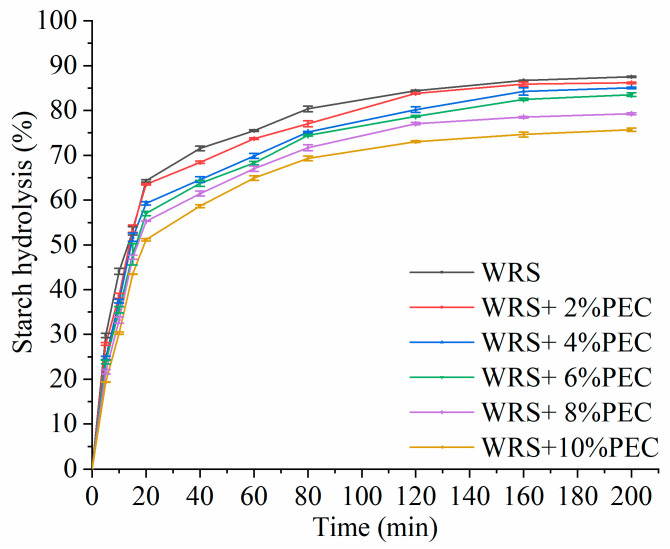
Starch hydrolysis curve of WRS−PEC mixed system.

**Table 1 foods-12-03981-t001:** Effect of PEC on retrogradation enthalpy of WRS during aging.

PEC Concentration	Δ*H_r_*(J/g)		
(%)	7 d	14 d	21 d
0	3.31 ± 0.01 ^a^	6.05 ± 0.03 ^a^	6.37 ± 0.02 ^a^
2	2.56 ± 0.01 ^b^	5.76 ± 0.02 ^b^	5.98 ± 0.06 ^b^
4	2.23 ± 0.13 ^c^	5.27 ± 0.01 ^c^	5.56 ± 0.07 ^c^
6	2.10 ± 0.01 ^d^	5.07 ± 0.02 ^d^	5.21 ± 0.08 ^d^
8	1.88 ± 0.01 ^e^	4.88 ± 0.02 ^e^	5.03 ± 0.04 ^e^
10	1.71 ± 0.01 ^f^	4.18 ± 0.02 ^f^	4.29 ± 0.03 ^f^

Mean value ± SD with different lowercase letters on shoulder tags in the same column indicate significant difference between them (*p* < 0.05).

**Table 2 foods-12-03981-t002:** The ratio of the absorbances (1047/1022) cm^−1^ and (1022/995) cm^−1^ in the infrared spectrum of WRS−PEC mixtures.

PEC Concentration (%)	(1047/1022) cm^−1^	(1022/995) cm^−1^
0	0.54 ± 0.01 ^a^	0.77 ± 0.01 ^e^
2	0.52 ± 0.00 ^b^	0.82 ± 0.01 ^d^
4	0.50 ± 0.01 ^c^	0.84 ± 0.01 ^d^
6	0.48 ± 0.00 ^c^	0.87 ± 0.01 ^c^
8	0.46 ± 0.01 ^d^	0.92 ± 0.02 ^b^
10	0.44 ± 0.01 ^e^	0.97 ± 0.01 ^a^

Mean value ± SD with different lowercase letters on shoulder tags in the same column indicate significant difference between them (*p* < 0.05).

**Table 3 foods-12-03981-t003:** The LF-NMR spectrum parameters of WRS−PEC mixed system during retrogradation.

Storage Time (d)	PEC Concentration (%)	T(ms)			A(%)		
		T_21_	T_22_	T_23_	A_21_	A_22_	A_23_
0	0	11.79 ± 0.93 ^a^	811.13 ± 0.00 ^a^	N.D.	1.9 ± 0.02 ^d^	98.1 ± 0.05 ^b^	N.D.
	2	10.26 ± 0.81 ^c^	740.69 ± 23.22 ^c^	N.D.	2.1 ± 0.01 ^c^	97.9 ± 0.24	N.D.
	6	8.92 ± 0.70 ^de^	644.22 ± 5.23 ^e^	N.D.	2.2 ± 0.01 ^b^	97.8 ± 0.21	N.D.
	10	8.11 ± 0.00 ^e^	613.59 ± 0.00 ^fg^	N.D.	2.4 ± 0.03 ^a^	97.6 ± 0.17	N.D.
7	0	10.73 ± 0.00 ^b^	775.91 ± 31.00 ^b^	N.D.	1.7 ± 0.01 ^e^	98.3 ± 0.13 ^a^	N.D.
	2	9.80 ± 0.81 ^c^	674.85 ± 53.05 ^e^	N.D.	2.0 ± 0.11 ^c^	98.0 ± 0.05 ^bc^	N.D.
	6	8.52 ± 0.70 ^de^	613.59 ± 0.00 ^fg^	N.D.	2.1 ± 0.05 ^c^	97.9 ± 0.03 ^c^	N.D.
	10	7.76 ± 0.11 ^f^	586.95 ± 46.12 ^g^	N.D.	2.2 ± 0.02 ^b^	97.8 ± 0.12 ^cd^	N.D.
14	0	9.33 ± 0.00 ^cd^	705.48 ± 0.00 ^d^	2154.43 ± 92.51 ^a^	1.4 ± 0.05 ^h^	98.1 ± 0.04 ^b^	0.4 ± 0.01 ^b^
	2	8.11 ± 0.00 ^e^	644.22 ± 43.06 ^e^	N.D.	1.8 ± 0.09 ^d^	98.3 ± 0.18 ^a^	N.D.
	6	7.06 ± 0.00 ^g^	560.31 ± 34.22 ^g^	N.D.	1.9 ± 0.03 ^d^	98.1 ± 0.22 ^ab^	N.D.
	10	6.14 ± 0.00 ^i^	533.67 ± 0.00 ^hi^	N.D.	2.2 ± 0.04 ^b^	98.0 ± 0.14 ^bc^	N.D.
21	0	6.75 ± 0.53 ^h^	613.59 ± 0.00 ^fg^	1873.82 ± 93.35 ^b^	1.1 ± 0.03 ^i^	98.0 ± 0.12 ^bc^	0.9 ± 0.03 ^a^
	2	6.44 ± 0.46 ^h^	510.50 ± 36.21 ^i^	1629.75 ± 98.96 ^c^	1.5 ± 0.02 ^g^	98.2 ± 0.02 ^b^	0.3 ± 0.04 ^c^
	6	5.34 ± 0.00 ^j^	464.16 ± 0.00 ^j^	N.D.	1.6 ± 0.11 ^f^	98.4 ± 0.12 ^a^	N.D.
	10	4.64 ± 0.00 ^k^	423.85 ± 30.36 ^k^	N.D.	1.8 ± 0.08 ^d^	98.2 ± 0.17 ^b^	N.D.

Mean value ± SD with different lowercase letters on shoulder tags in the same column indicate significant difference between them (*p* < 0.05).

**Table 4 foods-12-03981-t004:** Content of digestive component of WRS−PEC mixed system.

PEC Concentration (%)	RDS	SDS	RS
0	64.31 ± 0.26 ^a^	20.11 ± 0.35 ^b^	15.75 ± 0.09 ^e^
2	63.53 ± 0.14 ^a^	20.25 ± 0.05 ^b^	16.03 ± 0.1 ^e^
4	59.29 ± 0.35 ^b^	20.9 ± 0.26 ^ab^	19.81 ± 0.62 ^d^
6	57.04 ± 0.49 ^c^	21.65 ± 0.31 ^a^	21.31 ± 0.18 ^c^
8	54.81 ± 0.43 ^d^	21.78 ± 0.24 ^a^	23.4 ± 0.19 ^b^
10	51.61 ± 0.18 ^e^	21.87 ± 0.44 ^a^	26.52 ± 0.62 ^a^

Mean value ± SD with different lowercase letters on shoulder tags in the same column indicate significant difference between them (*p* < 0.05).

**Table 5 foods-12-03981-t005:** Starch hydrolysis parameters and *eGI* of WRS−PEC mixed system.

PEC Concentration (%)	*C_∞_*	*k*	R^2^	*eGI*
0	82.31 ± 1.82 ^a^	0.073 ± 0.006 ^a^	0.9776	85.25 ± 1.75 ^a^
2	80.88 ± 2.06 ^ab^	0.069 ± 0.007 ^ab^	0.9718	84.25 ± 1.96 ^ab^
4	78.54 ± 2.25 ^b^	0.065 ± 0.007 ^ab^	0.9694	82.73 ± 2.14 ^b^
6	77.51 ± 2.11 ^bc^	0.061 ± 0.006 ^ab^	0.9736	81.90 ± 1.98 ^bc^
8	74.75 ± 1.83 ^c^	0.060 ± 0.006 ^b^	0.9788	80.33 ± 1.69 ^c^
10	71.84 ± 1.52 ^c^	0.056 ± 0.004 ^b^	0.9850	78.47 ± 1.38 ^c^

Mean value ± SD with different lowercase letters on shoulder tags in the same column indicate significant difference between them (*p* < 0.05).

## Data Availability

Data available on request due to privacy.

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
