# Peer review of "Long-Term Retrogradation Properties and In Vitro Digestibility of Waxy Rice Starch Modified with Pectin"

_foods, 2023, doi:10.3390/foods12213981_

Round 1
Reviewer 1 Report
Comments and Suggestions for Authors
The manuscript describes the effect of addition of pectin on retrogradation characteristics and in vitro digestibility of waxy rice. It will be helpful for the researchers to develop low starch digestibility waxy rice with high satiety response. The authors tried their best for the compilation of the selected research content. However, it needs several corrections/ improvements as mentioned below.
• Brief background of conducted research area is not mentioned in abstract. Additionally, salient findings in quantifiable manner regarding modification in starch digestibility by addition of pectin should be mentioned.
• Authors describe that pectin addition with waxy rice reduces starch digestibility. However, in introduction section, only modification of corn starch with pectin is mentioned. How pectin improved low amylose content waxy rice is not clear. Please clarify and mention prior work related to improvement of rice starch digestibility by the addition of hydrocolloid compounds.
• Overall, introduction is shallow and needs additional information at several levels. For example, explain about long retrogradation and mention about mechanism of retrogradation and its impact on starch digestibility.
• In section 2.1, mention whether it is brown or milled rice flour.
• Line 107: I think it will be DSC instead of DSC3.
• Line 156: Elaborate the procedure of digestive juice composition and method.
• Line 161: Mention the amount and concentration of GOPOD taken.
Please check once throughout in the manuscript.
Reviewer 2 Report
Comments and Suggestions for Authors
Dear authors,
Overall, the manuscript is quite good. I have some recommendations for improvement.
Introduction
Line 48 should be better explained for first time authors on the question why it is necessary to "reduce digestibility and retrogradation" because retrogradation reduces the digestibility of starches.
Material and Methods
Line 161, the GOPOD kit reference of the test should be included.
G20, G120 and M0 should be described.
Results
Table 1, Table 2, Table 3, Table 4 and Table 5. It should be better described in the table which concentration is statistically significant.
Kind regards,
Reviewer 3 Report
Comments and Suggestions for Authors
The manuscript presents the use of pectin to slow down the retrogradation of starch from waxy rice during 21-days storage. The use of hydrocolloid to slow down starch retrogradation and digestibility has been widely studied and published. Pectin is also one of the common hydrocolloids to be used for this purpose. The results and discussion are clearly presented and in good structure of writing/content.
Other comments:
1. Introduction: Please state the novelty of this study clearly.
2. Waxy rice starch specification related to retrogradation should be reported (ie. %amylose and %amylopectin).
3. Specification of pectin should be reported (high or low methoxy pectin? DE=?).
4. Total starch of all samples (with/ without pectin) should be determined.
5. It will be more interesting to present the estimated glycemic index of all samples.
6. Line 353-354, "The addition of PEC significantly improved the digestibility of WRS."
What did authors mean "Improve the digestibility". According to the results, RDS was decreased while SDS was increased. The starch should be harder to digest. Please clarify the sentence.
Comments on the Quality of English LanguageOverall is fine.
Reviewer 4 Report
Comments and Suggestions for Authors
The manuscript is about the effects of pectin on waxy rice starch. I have comments given below:
Line 83: The rationalization of the study must depend on waxy rice starch, not starch generally. I also think that the objective of the study must be given in more detail. The objective should not be that "the effect of pectin on other starches have been studied, now we are studying its effect on waxy rice starch"
Line 181: It should be "gelatinized starch", not "gelatinization starch".
The authors gave the results but they just wrote one sentence about how the results can be used. I think how these result can be used part is very important and must be submitted in more detail. So the conclusion part seems weak.
The similarity report obtained from turnitin resulted 40% similarity without bibliographia. So, it must be below 20%.

The whole manuscript must be read carefully and corrected accordingly.
Round 2
Reviewer 2 Report
Comments and Suggestions for Authors
Dear Author,
Changes have been applied correctly.
Best regards,
Author Response
Dear Reviewer:
Thank you for your approval of our revised article!
Best wishes!